# Leadership – not followership – determines performance in ant teams

Thomas O. Richardson [1,2✉], Andrea Coti[1], Nathalie Stroeymeyt [1,2,3✉] & Laurent Keller [1,3]

Economic theory predicts that organisations achieve higher levels of productivity when tasks are divided among different subsets of workers. This prediction is based upon the expectation that individuals should perform best when they specialise upon a few tasks. However, in colonies of social insects evidence for a causal link between division of labour and performance is equivocal. To address this issue, we performed a targeted worker removal experiment to disrupt the normal allocation of workers to a cooperative team task – tandem running. During a tandem run a knowledgeable leader communicates the location of a new nest to a follower by physically guiding her there. The targeted removal of prominent leaders significantly reduced tandem performance, whereas removal of prominent followers had no effect. Furthermore, analyses of the experience of both participants in each tandem run revealed that tandem performance was influenced primarily by how consistently the leader acted as a leader when the need arose, but not by the consistency of the follower. Our study shows that performance in ant teams depends largely on whether or not a key role is filled by an experienced individual, and suggests that in animal teams, not all roles are equally important.

[1] Department of Ecology and Evolution, Biophore, University of Lausanne, Lausanne, Switzerland. [2] School of Biological Sciences, University of Bristol, Bristol, UK. [3] These authors contributed equally: Nathalie Stroeymeyt, Laurent Keller. ✉email: tom.richardson30@gmail.com; nathalie.stroeymeyt@gmail.com

The ecology of many terrestrial habitats is dominated by colonies of social insects (ants, bees, wasps & termites[1]), or by human settlements[2]. Human and insect societies often exhibit an extensive division of labour, whereby different groups of individuals focus upon different tasks for at least an intermediate period of time[3]. As theory predicts that it should increase group productivity[4,5], division of labour has often been suggested as the main driver of the ecological success of human and insect societies[6–12].

There is good evidence supporting a positive association between division of labour and productivity in humans. For example, novel age- and sex-based divisions of labour arising within human groups in the Upper Paleolithic are thought to have spurred an increase in group foraging efficiency, which may have allowed early humans to out-compete other archaic hominins[13]. Similarly, the great increase in productivity achieved during the Industrial Revolution was achieved through organisational changes in which generalist 'Jack of all trades' labour practices were superseded by specialisation, in which workers focus upon a more restricted task set[4]. However, although there is strong evidence for a link between division of labour and performance in humans and also in the minority of social insect species with polymorphic worker castes[14–18], there is little evidence for such a link the majority of species that have monomorphic workers.

Establishing a causal relationship between division of labour and task performance requires manipulative experiments to test whether disrupting the normal allocation of individuals to tasks negatively impacts upon their performance. Colonies of social insects represent an ideal study system to carry out such experiments, as the division of labour can be perturbed by manipulating the task demands and the labour supply can be easily monitored. However, so far, studies that have quantified task performance in social insects have been either purely descriptive[10,19,20] or have only manipulated task demands[21–23]. Further, the studies that did manipulate the labour supply focused on whether other workers switch tasks to compensate for the removal of their nestmates, and if so, which workers replace the removed individuals, but they did not measure how well the replacements performed the new tasks[24,25]. Here, we set out to test for a causal relationship between division of labour and task performance by simultaneously increasing the task demand and reducing the labour supply in order to disrupt normal task allocation. We focus upon a well-studied team task, that is, one in which there are multiple roles that each require the simultaneous cooperation of several individuals for successful completion[8], namely, tandem running in *Temnothorax* ants[26,27].

*Temnothorax* is a genus of monomorphic ants which nest in fragile natural cavities such as acorns and twigs. As these nests are vulnerable to damage and degradation, colonies are often faced with the considerable challenge of identifying a suitable new nest, and then emigrating there without splitting into fragments[28–30]. During the emigration, communication about potential new nest sites is organised by tandem running – stereotyped exchanges of tactile signals during which a knowledgeable ant (the leader) physically guides a naive nestmate (the follower) to a new nest site (Fig. 1a). As followers learn the location of the nest site to which they were led, and later recruit other ants back to the same site[20,31–34], the behaviour serves as a mechanism for disseminating valuable information via a decentralised communication network (Fig. 3a, Fig. S2). Two features of this system lend it to the study of the relationship between division of labour and task performance. First, recent work on tandem running in another species of *Temnothorax* revealed a division of labour between leaders and followers, with the leading task typically fulfilled by a specialist leader (i.e., an individual with a track record of leading rather than following), and the following task fulfilled by a specialist follower[20]. Second, task performance can be readily measured[32], as the tandem run is successful if the leader guides the follower all the way into the new nest, but unsuccessful if the pair lose contact with one another en route (Fig. 1b).

To evaluate whether disrupting the normal task allocation process affects tandem run performance, we carried out a series of targeted worker removals[25,35,36] where either prominent leaders, prominent followers, or both were removed from the colony. We then compared the performance of tandem runs between the treatments.

Finally, to tease apart the influence of leaders versus followers upon tandem run performance, we defined two measures of experience for each role, namely, activity and consistency. We then used multi-model selection[37–41] to quantify the associations between experience and performance, and to test for possible synergies between leading and following specialists.

## Results

**Division of labour within tandem teams.** Twelve colonies of *Temnothorax nylanderi* ants were induced to perform five successive binary choice nest emigrations. The first four emigrations

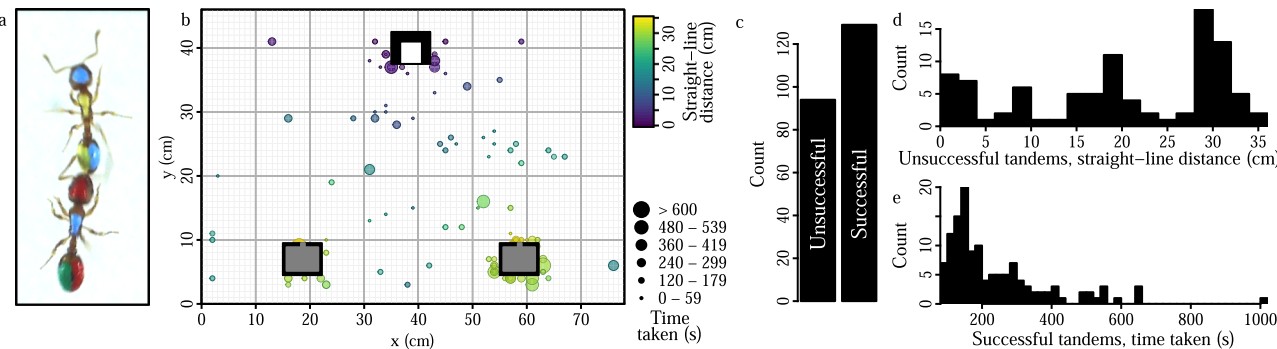

**Fig. 1 Quantifying tandem run performance in *Temnothorax nylanderi* ants. a** A tandem run between two workers each marked with a unique paint colour combination. The ant with paint code BYYB (Blue head, Yellow thorax, Yellow left gaster, Blue right gaster) leads, and ant RBGR (Red head, Blue thorax, Green left gaster, Red right gaster) follows. **b** The emigration arena. The initial low-quality nest is shown in white (top), and the two high-quality nests in grey (bottom). Points indicate the spatial distribution of the break-up locations of unsuccessful tandem runs. Successful tandem runs terminated inside a new nest, hence their end locations are not shown. **c** Breakdown of successful versus unsuccessful tandem runs observed during the fifth emigration. Histograms show the distribution of the straight-line-distances for unsuccessful tandem runs **d**, and the time taken by successful tandem runs **e**.

('baseline') were used to establish the task profiles for each individual in the colony. To induce the formation of atypical tandem runs, prominent tandem leaders and/or followers were selectively removed from the colonies just before the fifth 'test' emigration. The removals were split into four treatments: a prominent leader removal, a prominent follower removal, a combined prominent leader and follower removal, and a positive control where non-prominent leaders and followers were removed (see Methods for detail). In the test emigration, overall tandem performance was quantified using the proportion of tandem runs that successfully travelled all the way from the initial nest to the new nest without breaking up (success rate, 1c). The performance of successful tandem runs was further quantified by measuring the time taken to travel from the initial to the new nest (time taken, Fig. 1d). Additionally, as followers that participate in a tandem run that breaks up before reaching the goal may still acquire some information[34], the performance of unsuccessful tandem runs was further quantified by measuring the distance travelled toward the new nest before breaking up (straight-line-distance, Fig. 1e).

In total, 2143 tandem runs were recorded across 60 nest emigrations (five successive emigrations for each of 12 experimental colonies; median 35 tandem runs per emigration, range 13–66, Fig. S2). Participation in tandem running was skewed towards a minority of the colony population, with an average of only $32 \pm 1\%$ (SEM, $N = 48$ baseline emigrations) of the workforce engaging in at least one tandem run. This is in close agreement with previous work on *Temnothorax albipennis*[20,27] ($35 \pm 8\%$, $N = 12$, & $29 \pm 1\%$, $N = 30$), and *Temnothorax curvispinosus*[42] ($26 \pm 4\%$, $N = 6$). Considering leading and following separately revealed several differences between the two roles. First, leading was a more exclusive task than following, as $31 \pm 1\%$ of ants acted as follower at least once, whereas only $21 \pm 1\%$ of ants acted as leader at least once. Second, leading was more repeatable than following, as the number of tandem runs an individual led exhibited a significant positive correlation over emigrations separated by up to four weeks (i.e., between the first and the fourth baseline emigration, Fig. S4a, d), whereas the number of tandem runs an individual followed exhibited a weak positive correlation over only one week (i.e., between successive emigrations, Fig. S4b, e).

As in one other species of *Temnothorax*[20], we found an above-chance proportion of statistically significant negative correlations between the number of tandem runs an ant led in a given emigration and the number it followed in subsequent emigrations (Fig. S4c, f). This suggests that tandem leaders are specialised in the leading role. Furthermore, comparisons of the leading and following consistency of both participants within each tandem run revealed that leaders were typically more consistent in the leading role than their follower, and followers more consistent in the following role than their leader (Fig. S5a–c). Permutation tests confirmed that these consistency differences between leaders and followers were greater than expected by chance (Fig. S5d). Hence, pairs of tandem running *T. nylanderi* ants are not randomly assembled, but rather their composition is consistent with a stable division of labour between specialised leaders and followers.

Although individual workers did switch within emigrations from following to leading (overall $P(F \rightarrow L) = 0.57$) and from leading to following (overall $P(L \rightarrow F) = 0.16$), the most specialised workers exhibited distinctly asymmetrical task switching preferences. Thus, 'consistent leaders' – workers that led at least once in every baseline emigration – were more likely to switch to leading after having followed (Fig. 2a, Linear Mixed Model (LMM), $P(F \rightarrow L) \sim C_L$, $\chi^2 = 26$, d.f. $= 1$, $p < 0.0001$), but were no more or less likely to switch to following after having led (Fig. 2c, $P(L \rightarrow F) \sim C_L$, $\chi^2 = 1.2$, d.f. $= 1$, $p = 0.27$). Similarly, 'consistent

followers' were more likely to switch to following after having led (Fig. 2d, LMM, $P(L \rightarrow F) \sim C_F$, $\chi^2 = 14$, d.f. $= 1$, $p < 0.0001$), but were no more or less likely to switch to leading after having followed (Fig. 2b, $P(F \rightarrow L) \sim C_F$, $\chi^2 = 1.2$, d.f. $= 1$, $p = 0.27$). Permutation tests further demonstrated that all of the observed associations between switching probability and consistency were significantly different to those expected by chance alone ($p \leq 0.024$ in all permutation tests; Fig. S6). These analyses suggest that despite the exchange of workers between the leading and following tasks during an emigration, the division of labour between leading and following specialists is maintained by asymmetric switching preferences.

**Tandem performance is disrupted by targeted removal of prominent leaders.** To test whether the targeted removal treatments were effective in manipulating the composition of tandem runs, we compared the total leading and following activity, and the leading and following consistency of the tandem participants in all four treatments. Removing the prominent leaders resulted in tandem runs in which the leaders had significantly lower leading consistency, and had been significantly less active in leading than in the baseline emigrations (Fig. 3b, Fig. S7a, c). Similarly, removing prominent followers resulted in tandem runs in which the followers had significantly lower following consistency, and were significantly less active in following than in the baseline emigrations (Fig. 3b, Fig. S7b, d). Hence, the targeted removal treatments succeeded in modifying tandem run composition.

We next conducted colony-level analyses to test whether tandem run performance was affected by the four targeted removal treatments. In spite of the low sample size of three colonies per treatment, two of the three measures of tandem performance exhibited significant differences between the removal treatments (GLMM, tandem run success rate ~ treatment, $\chi^2 = 11.0$, d.f. $= 3$, $p = 0.013$; LMM, straight-line-distance among unsuccessful tandem runs, ~ treatment, $\chi^2 = 12.0$, d.f. $= 3$, $p = 0.0073$; LMM, time taken by successful tandem runs, ~ treatment, $\chi^2 = 7.2$, d.f. $= 3$, $p = 0.066$). Pairwise contrasts between treatments revealed that the only treatments that induced a significant reduction in tandem run performance relative to the control were those in which prominent leaders were removed (Fig. 3c, d). This suggests that prominent leaders are crucial for tandem run performance, but prominent followers are not. Surprisingly, there was no evidence of synergy between prominent leaders and followers, as the simultaneous removal of both prominent leaders and followers did not reduce performance more than when only prominent leaders or only prominent followers were removed (Fig. 3b, c).

**Tandem performance depends upon the long-term consistency of the leader.** Although the targeted removal treatments were successful in generating tandem runs with marked differences in the activity and consistency of the leader and follower in their respective roles, there was still considerable overlap between the composition of tandem runs in each treatment (Fig. S7). To tease apart the influence of the leader's experience (measured as either its past consistency in leading across emigrations, $C_L$, or the total number of tandem runs it previously led, $A_L$), the follower's experience (measured as either its past consistency in following across emigration, $C_F$, or the total number of tandem runs it previously followed, $A_F$, and tandem run timing (rank of a tandem run within a given emigration) upon tandem run performance, we pooled the data from all four treatments and carried out an individual-level analysis using multi-model selection[37–41].

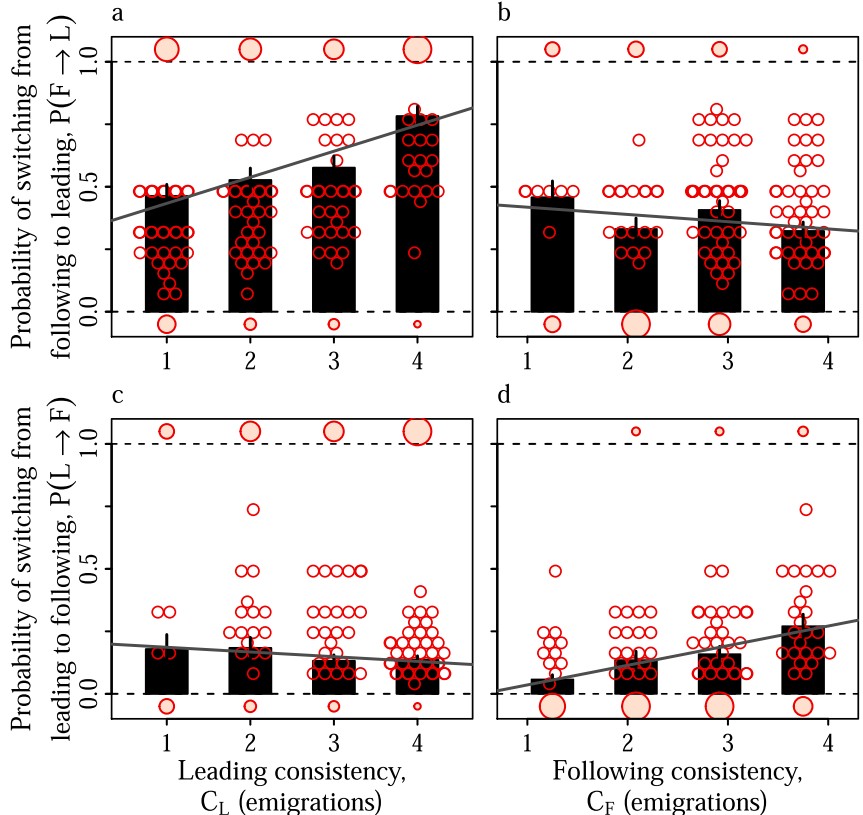

**Fig. 2 Asymmetric switching between leading and following.** Bars and errors represent the mean and standard error of the per-tandem switching probabilities, averaged across all individuals with a given leading or following consistency. **a**, **b** Per-tandem probability that an ant acting as a follower in a given tandem switches to leading in the next tandem. **c**, **d** Per-tandem probability that an ant acting as a leader in a given tandem switches to following in the next tandem. Grey lines show the LMM fits mentioned in the text. Empty points represents the per-tandem switching probability for a single individual. Filled points represent individuals with switching probabilities of zero or one. The diameters of the filled points are proportional to the number of ants that had a switching probability of zero or one.

This analysis revealed that the timing of tandem runs within an emigration had little influence on any of the three tandem performance metrics (Fig. 4d–f). Instead, the consistency of the leader in the leading task $C_L$ was the most important predictor of both the overall tandem success rate (Fig. 4a) and also the straight-line distance covered by unsuccessful tandem runs (Fig. 4b, Table S2). Furthermore, the model-averaged coefficient associated with the leader consistency was significantly greater than zero for the tandem success rate (Fig. 4d), and the straight-line-distance covered by unsuccessful tandem runs (Fig. 4e). Although the consistency of the leader in the leading task was one of the most important predictors of the third performance metric (i.e., the time that successful tandem runs took to reach the new nest), the coefficient was not significantly different from zero (Fig. 4c, f). Similarly, both of the interaction terms ($C_L \times C_F$ & $A_L \times A_F$) had very low importance (i.e., they appeared in few of the 95% confidence set models) and near-zero coefficients, which again suggests the absence of synergistic interactions between specialist leaders and followers.

An analysis of redundancies between the model predictors revealed only weak collinearity in the models in which the response was the overall tandem success rate or the straight-line distance covered by unsuccessful tandem runs (Fig. S8, success rate models; mean & standard of $|\bar{r}| = 0.34 \pm 0.02$, variance inflation factor, VIF $= 4.1 \pm 0.04$, straight-line-distance models; $|\bar{r}| = 0.32 \pm 0.04$, VIF $= 2.6 \pm 0.3$). The predictor collinearity was slightly higher in the models for the time taken by successful tandem runs ($|\bar{r}| = 0.38 \pm 0.02$, VIF $= 6.2 \pm 0.6$), which might

explain why none of the predictors were significant. Filtering the confidence set models to eliminate redundancies caused by inclusion of slightly collinear terms did not did not change our results, as in these filtered models the model-averaged coefficient for leader consistency $C_L$ was again significantly greater than zero for both the success rate and also the straight-line distance performance metrics (Fig. S9).

## Discussion

Historically, most work on division of labour in social insects has focused on the most conspicuous examples, such as the division of labour between reproductive and sterile worker castes[43], between polymorphic worker castes[5], and between 'temporal castes' of younger, inside-nest and older, outside-nest workers[7]. However, although difficult to identify, more subtle divisions of labour appear also to be ubiquitous. For example, among nurses of the Pharoah ant, there is a further subdivision between nurses that specialise on young larvae and nurses that specialise on feeding older larvae[44]. Similarly, among the outside-nest workers of the honeybee, foraging is subdivided between scouts that explore for new resources, and recruits that exploit them[45]. Our results indicate that tandem running may represent another subtle division of labour among outside-nest workers.

Previous work on *Temnothorax albipennis* ants suggested that tandem runs exhibit a stable division of labour between leading and following specialists[20]. By repeatedly challenging entire colonies to emigrate into a new nest, we confirmed the presence

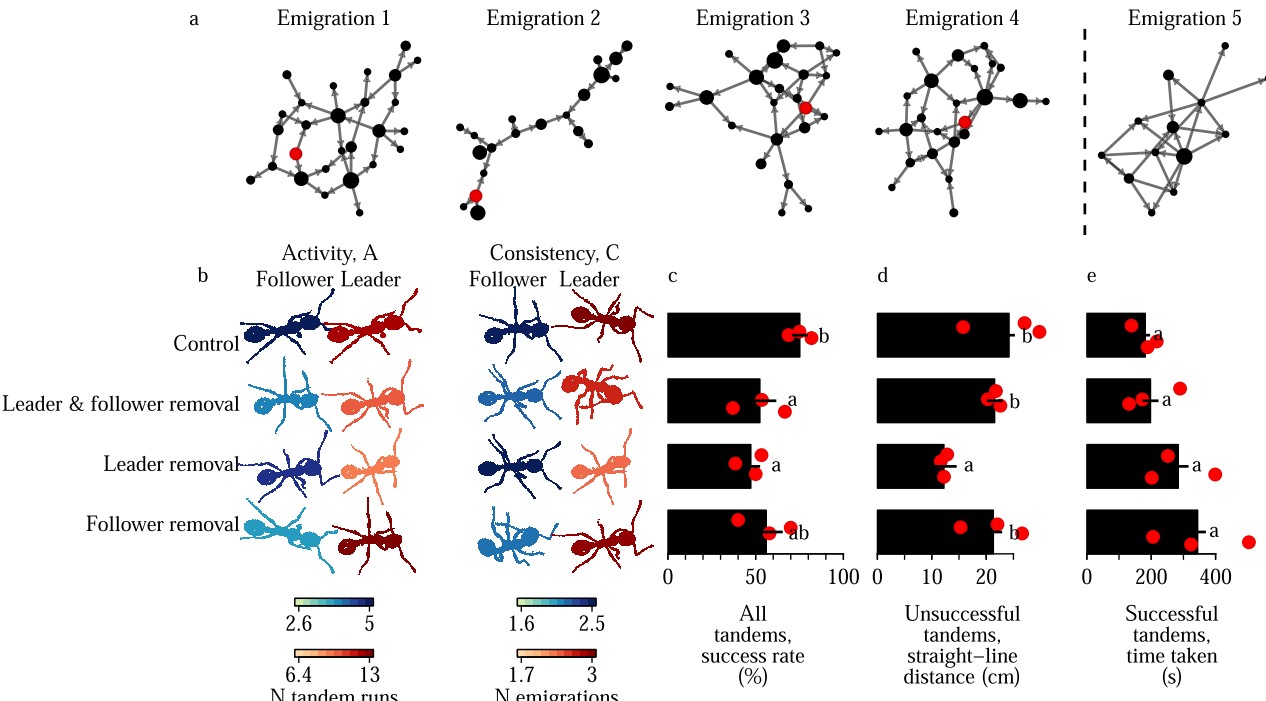

**Fig. 3 Targeted worker removals show that tandem performance is most disrupted by removal of prominent leaders. a** Tandem recruitment networks for one colony in the leader removal treatment. Links represent tandem runs, and are directed from the leader to the follower. Red node; a prominent leader with leading consistency $C_L = 4$ emigrations, and leading experience $A_L = 12$ tandem runs, which was removed between emigrations four and five. Node size indicates the number of tandem runs each ant led across the four baseline emigrations. **b** The effect of the targeted removals upon the composition of tandem pairs. Colours indicate the mean activity or consistency for leaders and followers in a given treatment. **c** The proportion of tandem runs that successfully reach one of the high-quality nests without breaking up. **d** The distance that unsuccessful tandem runs cover towards one of the high-quality nests before breaking up. **e** The time taken for successful tandem runs to travel all the way to one of the new nests. Bars & errors represent the mean & standard errors, where the means are grand means calculated from the colony means (red points). Letters indicate post-hoc contrasts from the colony-level mixed-models.

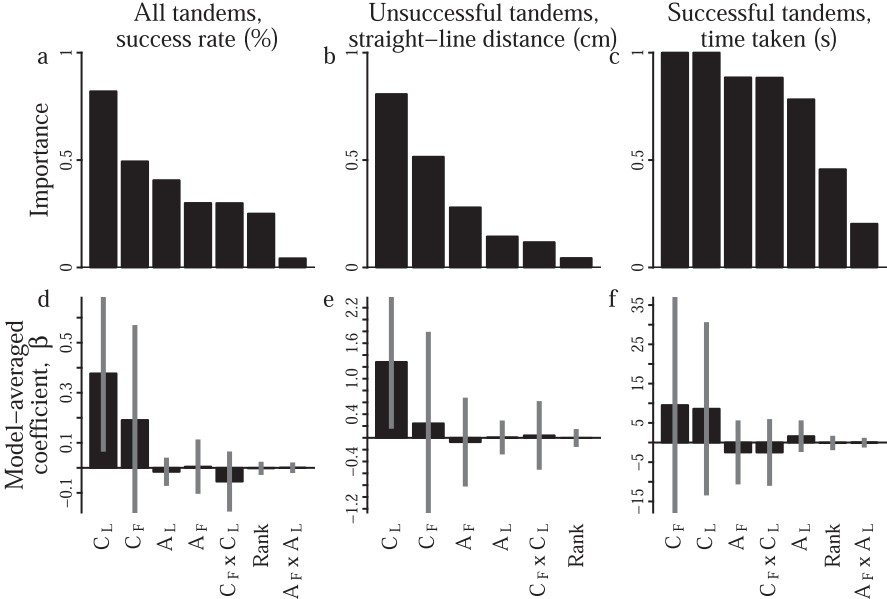

**Fig. 4 Model averaging shows that tandem performance depends chiefly upon the number of baseline emigrations in which the leader acted as a leader, $C_L$. a–c** Relative predictor importance, and **d–f** model-averaged coefficients across all models in the 95% confidence set. Bars & errors represent coefficient means & standard errors for each predictor; C consistency, A tandem activity. Subscripts indicate the task. `Rank' indicates the within-emigration tandem ordering.

of a very similar division of labour between leading and following specialists during tandem running in *Temnothorax nylanderi*. Our targeted removals of prominent leaders and followers also suggested that the performance of a pair of tandem running ants is determined primarily by whether or not the leading role is fulfilled by a leading specialist: whereas the removal of prominent leaders induced significant performance reductions, performance was not significantly reduced by the removal of prominent followers.

The absence of significant performance reductions associated with the removal of prominent followers suggests that in team tasks – those with multiple roles that require the simultaneous cooperation of several individuals for successful completion[8] – not all roles are equally important. Interestingly, field studies of cooperative hunting in mammals[46–48], collective navigation in birds[49,50], and cooperation within sports teams[51], suggest that some roles are more important than others, and that in order for the group to succeed it is crucial that specialised or experienced individuals fulfil certain key roles. Our study provides evidence that team performance in social invertebrates also depends on an appropriate matching of specialists to key roles within the team.

One possible explanation for the greater influence of tandem leaders is that leadership may require a more cognitively demanding set of skills than followership. For example, as well as carrying out the navigation duties, tandem leaders must also evaluate the tactile feedback provided by the follower, and decide how long to wait for the follower after the pair lose contact with one another[32]. Indeed, in other *Temnothorax* species, tandem leaders and followers exhibit distinct gene expression profiles, with leaders displaying significantly greater expression levels for genes associated with olfactory, visual and associative learning compared to followers[52]. The additional cognitive demands associated with leading might thus act as a filter such that able individuals become over-represented among the leaders. This filtering hypothesis is supported by the finding that in *Temnothorax* ants leading is a more exclusive task than following[20,27,42]. Alternatively, since it is the leader that determines the route taken by the tandem run, it is also possible that any effects of follower expertise on tandem performance were masked by the outsized influence of the leader.

Given the large body of work supporting a link between experience and performance in social insects[19,21,53–61], it is natural to expect that tandem runs should perform better when the role performed by the participants is appropriately matched to their previous activity within the role, such as when the leader is an ant which previously acted as a leader in many tandem runs. However, our individual-level analyses did not support this; the total number of tandem runs that an ant had led during the baseline emigrations (i.e., its total leading activity) was not an important predictor of the performance of the tandem runs it led in the fifth emigration. Rather, tandem run performance exhibited a positive dependence upon the number of baseline emigrations in which an individual acted as a leader at least once (i.e., by leading consistency). In other words, the tandem runs led by consistent and active leaders were no better than those led by consistent but less-active leaders. This surprising result has parallels across a wide range of human endeavours, such as professional sports and music, where 'spaced repetition' – doing a little often – has been shown to result in higher performance than 'cramming' in which repetitive practice is concentrated into a short time period[62].

What makes a good leader? Across a range of species, leadership has been shown to be associated with differences in knowledge[63], certainty[64], motivation[65], speed[66], spatial position[67], and age[68]. However, the proximate mechanisms that generate these differences are still poorly understood[69]. Such differences can stem from congenital task preferences. For example, honeybee scouts (leaders) and recruits (followers) exhibit different predispositions for associative learning[70], and in desert ants, workers belonging to different patrilines tend to specialise upon different tasks[71]. Differences in the tendency to lead might also stem from behavioural processes that promote positive feedback, such as reinforcement learning and sensitisation[17]. For example, workers of the clonal raider ant that successfully complete a foraging trip become more likely to forage again, whereas those that fail to do so become less likely to forage again[57]. Clearly, more work is required to disentangle the contributions of innate predispositions versus life experience, and to better address the old claim that 'great leaders are born, not made'.

## Methods

**Colony collection and housing**. Twelve colonies of *Temnothorax nylanderi* consisting of one queen, adult workers and developing brood were collected in the Forêt de Dorigny, Switzerland, between June and October 2018. The study was organised into three replicates, each involving four colonies. All colonies containing between 71 and 116 workers (Table S1). All four colonies within each replicate were collected exactly three days prior to the onset of the experimental procedure (i.e., paint-marking, see below for detail). See 'Colony collection & maintenance' in the Supplementary Information.

After collection, colonies were transferred to artificial nests made of a cardboard perimeter sandwiched between two glass slides, and placed in $11 \times 15 \times 3$ cm holding boxes with Fluon-coated walls. These initial nests contained a single small, low-ceiling chamber ($25 \times 30 \times 1$ mm internal dimensions), with a transparent ceiling to allow light to enter, and a detachable front wall whose removal allowed increasing the width of the entrance from 2 to 30 mm. *Temnothorax* ants behave as though such nests are of poor quality[72].

**Applying unique paint codes to ant workers**. Three days after collection, every ant in each colony was marked with a unique paint code. These codes consisted of unique combinations of four different body locations (head, thorax, left & right gaster), and six colours (R: red, G: green, Y: yellow, W: white, B: blue and P: pink, Fig. 1a, Fig. S1). The paint-marking scheme was designed to generate a desired number of colour-code combinations that were maximally redundant, allowing the identity of an ant to be established even after the loss of up to two paint marks (R code freely available, https://zenodo.org/record/3240545). See the section, 'Applying unique paint codes to ant workers' in the Supplementary Information.

**Colony emigrations**. Each colony was subjected to five emigrations, with a one week interval between successive emigrations. The objective of the first four emigrations ('baseline emigrations') was to quantify the specialisation of each worker in tandem leading and tandem following, in order to identify a list of target ants to remove prior to the fifth emigration. The objective of the fifth emigration ('test emigration') was to test whether targeted removals of prominent leaders and/or followers influenced tandem quality.

On the morning of each emigration, a colony housed in a low-quality initial nest was placed in a large rectangular experimental arena ($46 \times 78$ cm) in a triangle arrangement with two identical empty new nests, whose entrances were equidistant to that of the initial nest (Fig. 1b). The two new nests had four times the volume of the initial nest, ($50 \times 30 \times 2$ mm), a single 2 mm wide entrance, and were covered with a cardboard ceiling so that their interiors were dark. *Temnothorax* ant colonies consistently prefer such 'high quality' nests over smaller, brighter alternatives[72] and spontaneously emigrate to these nests when housed in low-quality nests ('move to improve' behaviour[73]).

In order to trigger an emigration, the quality of the nest was further reduced by removing the detachable front wall immediately after placing the initial nest into the arena. This design was chosen because the quality difference between the initial and new nests motivates nest-site discoverers to lead tandem runs, whilst the binary choice between identical options delays the formation of a consensus, leading to large numbers of tandem runs[20,73].

During each emigration a handheld digital microscope was used to identify the paint codes of both the leader and the follower in each tandem. The participation records gathered during the baseline emigrations were used to quantify leading and following consistency and activity for each individual prior to the fifth emigration. Additionally, a digital video camera (Sony HDRCX240) placed above the arena recorded the entire emigration (mean & S.E. of time from emigration start to removal of the last brood item from the initial nest; $261 \pm 27$ min). In the fifth emigration, these high-definition videos were used to quantify task performance by determining whether each tandem took to reach a nest, and if it failed to do so, exactly where in the arena it broke up. The audio track of the overhead camera was also used to record a running commentary on the emigration, including follower switches (i.e., occasions when one tandem follower was replaced by another ant), and the nest origin of each tandem. The nest origin was used to distinguish between 'forward' tandem runs (leading from the old nest to the new nest) and 'reverse' tandem runs (leading from the new nest to the old nest[74]).

After the ants removed the last brood item from the initial nest, all ants and nests were removed from the arena, and the arena floor was scrubbed with 70% ethanol to remove any pheromones laid by the ants. The colony was then moved into a new low-quality nest and placed back in its storage box for one week until the next emigration.

**Measuring role consistency and activity**. As tandem running by individual ants was unevenly distributed across the four baseline emigrations, we defined two complementary measures of experience. First, 'consistency' was assessed by counting the number of baseline emigrations in which each individual played a given role in at least one tandem run[20]. Thus, leading consistency, $C_L$, was assessed by counting the number of baseline emigrations in which an individual acted as a leader in at least one tandem run[20]. The consistency of the follower in the following role, $C_F$, was defined equivalently.

Second, 'activity' was defined by the number of tandem runs in which an individual played a given role across all baseline emigrations. Thus, leading activity, $A_L$, was defined as the total number of tandem runs in which an ant acted as a leader during the four baseline emigrations. Following activity, $A_F$, was defined equivalently.

**Targeted removal of prominent leaders and followers**. The tandem running records collected during the four baseline emigrations were used to identify prominent leaders and followers. The removals were conducted one day before the fifth emigration (Fig. S2). To identify prominent leaders and followers we used a hierarchical ranking procedure to assign two ranks to each individual, one for the leading and one for the following task. For the leading task, all ants were first ranked according to their leading consistency $C_L$ (Fig. S3). Ties were broken by ranking tied ants according to their leading activity, $A_L$. Ranks for the following task were assigned in the same way.

Each colony was then randomly assigned to one of four targeted removal treatments. The objective of these removals was to induce the formation of non-standard tandem pairs, for example a leader that did not typically act as a leader and a follower that did not typically act as a follower. As colonies varied in the size of their worker populations, and in the proportion of the workers that engaged in tandem running, colonies were subjected to a standardised removal of 50% of the 'tandem runners' (i.e., individuals that had participated in tandem running at least once during the baseline emigrations).

To induce the formation of tandem runs in which the leader had not previously been a prominent leader, but the follower had been a prominent follower, we removed the top-ranked 25% of tandem leaders and the bottom-ranked 25% followers ('prominent leader removal', Fig. S3). Conversely, to induce tandem runs in which the leader had previously been a prominent leader, but the follower had not been a prominent follower, we removed the top-ranked 25% of the tandem followers and the bottom-ranked 25% of the tandem leaders ('prominent follower removal' treatment). To induce tandem runs in which neither the leader nor the follower had previously been prominent in their respective roles, we removed the top-ranked 25% tandem leaders, and the top-ranked 25% followers ('prominent leader & follower removal' treatment). Finally, to induce tandem runs in which both the leader and the follower had previously been both prominent in their respective roles, we removed the bottom-ranked 25% tandem leaders, and the bottom-ranked 25% followers. This treatment was used as a control for the disruption caused by the removal procedure, while maintaining a more standard tandem pair composition (i.e., experienced leaders leading experienced followers[20]).

**Statistics and reproducibility**
*Quantifying tandem run performance*. We used three measures to quantify the performances of all the tandem runs from the fifth 'test' emigration. First, the 'success rate' was defined by the outcome of each tandem run. Tandem runs that progressed all the way from the old nest to one of the new nests were defined as successful, whereas those that broke up before reaching one of the new nests were defined as having failed. The 'straight-line-distance' was defined for unccessful tandem runs only, and was obtained by subtracting the beeline distance between the tandem run break-up location and the entrance of the nearest new nest, $l$, from the distance between the entrance of the initial nest and either new nest, that is $35\ cm - l$ (Fig. 1b). Finally, the 'time taken' was defined for successful tandem runs only, and was defined as the interval between time when the tandem run left the initial nest, and the time that it broke apart.

Reverse tandem runs (those that originate from one of the high-quality nests and proceed towards the initial nest) are more unstable than forwards tandem runs[28], and their role in colony emigration is not fully understood[74]. Similarly, the effect of follower switching on tandem run performance is unknown, and switching events make it difficult to assign the correct degree of follower experience and specialisation to a given tandem run. Therefore, analyses of tandem performance excluded all reverse tandem runs and all tandem runs involving follower switches.

*Task switching*. Sequences of leading and following acts were used to calculate the per-tandem task switching probabilities across all baseline emigrations. Thus, an individual that followed once and then led three times in the first baseline emigration (sequence; F,L,L,L), followed thrice then led once time in the second baseline emigration (F,F,L), but was not involved in the third or fourth baseline emigrations, had a probability of switching from following to leading, $P(F \rightarrow L) = 2/3$, as two of the three tandems in which it acted as a follower were succeeded by a tandem in which it acted as the leader. Similarly, this individual's probability of switching from leading to following was $P(L \rightarrow F) = 0$, as none of the tandem runs in which it acted as a leader were succeeded by a tandem in which it acted as a follower.

Possible confounding associations between the switching probabilities and both $C_L$ and $C_F$, were investigated using data permutations. See the section, 'Permutation tests for task switching' in the Supplementary Information.

*Mixed-effects modelling*. All statistical comparisons were conducted with mixed-effects models. A first set of mixed models made colony-level performance comparisons between treatments. As these were colony-level analyses, the responses were within-colony means, calculated by averaging the performance of all tandem runs produced by a given colony during the fifth emigration. One model was specified for each of the three performance metrics (tandem run success rate, straight-line-distance among unsuccessful tandem runs, and time taken by successful tandem runs).

A second set of individual-level mixed models analysed the relationships between tandem performance and role consistency and activity. In these models, the response was the performance of a given tandem run, and the main effects were the consistency and activity of the leader in the leading role ($C_L$, $A_L$), and the consistency and activity of the follower in the following role ($C_L$, $A_L$). To detect synergies between leaders and followers we included a leader-follower interaction term for consistency, $C_L \times C_F$, and for activity, $A_L \times A_F$. To test for an effect of the timing of a tandem run within an emigration, these models also included the rank of each tandem run (1st, 2nd,..., $n$th) as a further main effect. Furthermore, as the analyses of tandem run performance were based upon individual-level data in which a single ant may have led (or followed) multiple tandem runs, the models included the identity of the leader and follower as random effects. As the same paint codes were applied to ants in different colonies, ant identity was nested within colony identity (e.g. C4_RBGR), which was itself nested within replicate number (A,B,C).

To quantify the importance of the different predictors, whilst also avoiding the pitfalls associated with stepwise model selection procedures, the individual-level mixed models were filtered using an information-theoretic approach known as 'model selection and multimodel inference'[37,38]. This method takes the global (i.e., the most complex) model as an input, fits all possible subset models, and ranks them according to their Akaike Information Criterion. After ranking, a 95% confidence set of the most parsimonious models is generated by eliminating the lowest-ranked models. The confidence set is used to calculate the 'relative importance' of each predictor: those that appear in most models or in the high-ranked models have an importance approaching 1, whereas predictors that appeared in only a few models or in the low-ranked models have an importance approaching 0. See the section, 'Multimodel inference and model averaging' in the Supplementary Information.

Models predicting the overall success rate were specified as generalised linear mixed models (GLMM) with a binomial error distribution and a logit link function, whereas models predicting both the straight-line-distance and time taken were specified as linear mixed models (LMM). Conformity to model assumptions was assessed by measuring the residual skewness and kurtosis, and by visual inspection of residual quantile plots. Post-hoc contrasts between treatments were carried out with the Benjamini–Hochberg correction for multiple comparisons.

**Reporting summary**. Further information on research design is available in the Nature Research Reporting Summary linked to this article.

## Data availability
The data analysed in this paper are freely available (https://zenodo.org/record/3234428).

## Code availability
Computer code for producing redundant paint mark combinations is freely available (https://zenodo.org/record/3240545).

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

## Acknowledgements

L.K. acknowledges funding by the Swiss NSF and the European Research Council (ERC Advanced Grant 'resiliANT', no. 741491). N.S. acknowledges funding by the Swiss NSF (Eccellenza Professorial Fellowship PCEFP3_181209) and the European Research Council (ERC Starting Grant 'DISEASE', no. 802628). The authors thank T. Kay, E. Frank & M. Rüegg for detailed feedback on the manuscript.

## Author contributions

T.O.R. & N.S. conceived the study. A.C., N.S., T.O.R. & L.K. designed the experiments. A.C. performed the experiments. T.O.R., N.S. & A.C. analysed the data. T.O.R., N.S. & L.K. supervised A.C. T.O.R., N.S., A.C. & L.K. wrote the paper.

## Competing interests

The authors declare no competing interests.
