## [Peer Review File · Communications Biology]

Reviewers' comments:

Reviewer #1 (Remarks to the Author):

Manuscript Number: COMMSBIO-20-2367

Title: Leadership -- not followership -- determines performance in ant teams

The study used tandem running in *Temnothorax* ants to investigate division of labour in ants. This approach allowed to measure performance (successful relocation, distance to target nest) and to compare it between groups in which either the most active leaders, followers, both or hardly specialised ants were removed. To do so, a baseline for each ant was assessed in 4 emigration trials, before the manipulated 5th trial took place.

The study was well done, the manuscript is well written and clearly conveys the hypothesis and the used procedures. The statistical analysis is thorough and well documented. All in all, I have found no major issues and approve the publication in *Communications Biology* after the questions below have been addressed:

- The two measurements, success rate and distance covered, are not independent. All successful relocations always have a distance of 35cm. I would assume that the skewed distributions (many 35s; as shown in figure 1c) led to a violation of normality of residuals required for the LMM. Can the authors disprove that claim?
- Distance covered suggests that this is an estimate of how many centimetres the ants travelled. However, any sinuosity of the path is not reflected in the measurement. Can the authors approve that a straight line between the nests is a good approximation of the travelled route? Otherwise, the name of the metric is misleading.
 - o Moreover, the measures established in the manuscript might not capture follower performance in its entirety. Poor followers might cause detours or longer breaks. This might be reflected by the time taken per trip. Can the authors provide any insights on travel times?
- While we would expect a leader ant to do multiple trips per trial, it is interesting that also some followers do multiple runs within one trial. Probable explanations for that behaviour could be added to the discussion from line 162. In case that followers also have important functions such as e.g. cross-comparisons of nests known by previous research, it should be added as information to the introduction.
- The authors included both S_{lead} and S_{follow} in models during the model selection procedure. However, to my understanding, $S_{lead} = 1 - S_{follow}$. This information would thus be redundant, and these predictors are not independent. I am thus unsure about the validity of such a model. Can the authors exclude an effect on the model selection process? E.g. S_{lead} was not always in the model before S_{follow} was added.

Minor

- In Table 1 I would suggest to sort by species
- In line 95, MSE refers to mean squared error or SEM?
- I would suggest to add a small table or inset to figure 2 where the composition/procedure for the four groups is listed (lines 290 – 300)
- Very long sentence starting at l178, split.
- Line 337, so here no removal of reversal runs and switches took place?
- I would suggest to add the average baseline counts for the variables specialisation, emigration experience and tandem experience in the main text, so people do not have to extract them from figure S6.
- Line 56 exchange 'brunt' with 'most' to make the manuscript more accessible
- L75 lost contact
- Fig1 the corresponding randomly-selected
- L179 because of the high degree
- L279,80 sentence is confusing, maybe write: two ranks, one for the leading and one for the following task for each individual
- L319, shouldn't it be replicate nested in colony identity?

Reviewer #2 (Remarks to the Author):

Division of labor is often mentioned as one of the factors (among others) responsible for the ecological success of ants. Increased task performance at the colony level usually outcomes from a higher tasks' achievement by specialized individuals.

This research work is organized around two complementary axes.

The first one aims to specify the relative importance of the ants specialized in the leading and in the following on the effectiveness of tandem runs . The second axis aims to understand through statistical models whether task specialization or experience of the observed individuals are the main factors influencing the performance of tandem runs.

The paper is very well-written, figures in the main paper are informative and supported by many useful tables/figures in the supplementary.

Here follow my main comments on the different sections of this paper

Introduction

The authors underline that a positive association between specialization and task performance is relatively little supported by existing literature. I agree with them as far as experimental work on monomorphic workers is concerned. This is far less true when considering the many papers that demonstrated the link between task performance and specialization in polymorphic ants as well as by the theoretical studies made on this topics.

Here follow some suggested references (far to be an exhaustive list...)

-Wilson EO. Caste and division of labor in leaf-cutter ants (Hymenoptera: Formicidae: Atta). *Behav Ecol Sociobiol.* 1983; 14:

-Wilson EO. The relation between caste ratios and division of labor in the ant genus *Pheidole*. *Behav Ecol Sociobiol.* 1984

-Detrain, C., & Pasteels, J. M. 1991. Caste differences in behavioral thresholds as a basis for polyethism during food recruitment in the ant, *Pheidole pallidula* J. *Insect behave.* 4(2)

-Mirenda JT, Vinson SB. Division of labour and specification of castes in the red imported fire ant *Solenopsis invicta* Buren. *Anim Behav.* 1981; 29.

- Johnson BR. Organization of work in the honeybee: a compromise between division of labour and behavioural flexibility. *Proc R Soc Lond B Biol Sci.* 2003

-Theraulaz, G., Bonabeau, E., & Deneubourg, J. N. (1998). Response threshold reinforcements and division of labour in insect societies. *Proc Royal Soc London. Series B: Biological Sciences,* 265(1393).

-Also many studies on Army ants

To illustrate division of labor, the authors chose to study cooperative tandem runs during nest emigration in *Temnothorax* ants. I wonder if the leading "task" and the follower "task" can be considered as a division of labor similar to that observed between e.g. nest cleaners, nurses or foragers. This appears to me rather as a single task based on a communication comparable to that of other recruitments (by group or by trail).

At least, the authors should comment on this and widen their conclusions/discussion on the influence of leaders (scouts) VS followers in group and trail recruitment.

Removal Experiments

The first section reports experiments assessing how much the removal of either top-specialized leaders or followers will influence the performance of tandem runs. The authors found that the removal of specialized leaders had more impact than that of followers as shown by figure 2.

I am convinced about the influence of prominent leaders on the quality of tandem runs. However, the sample size associated to graph bars and statistical tests is not given. The figure caption specifies that the average values were calculated on all tandem runs but the sample size should be

indicated over each bar.

Furthermore, I think that it is not correct to average values measured on the individual tandem runs. As a rule, if one wants to evaluate the impact of a manipulation (here the removal of a subset of workers out of a given colony), the measure of this impact should also be done at the colony level. This means that tandem run performance should be done for each of the tested colony (e.g. the average distance of all tandem runs performed during one nest emigration by a given colony). This also means that bars shown on figure should be averaged on a sample of only $n=3$ (knowing that each of the 12 colonies is assigned to one of four conditions for the 5th nest emigration).

Such a sample size is definitely too small and needs to be increased to check whether the reported differences would be maintained. I am well aware that this requires additional experiments but, on the other hand, such a colony-level analysis will provide additional interesting data such as the possibility to compare the dynamics of tandem runs along the emigration process across the different conditions of workers' removal.

Specialization VS Experience

The second section aims to disentangle the effects of experience and specialization on the performance of tandem runs.

At line 79, the authors state that they "capitalized upon the observation that specialisation and experience are only loosely coupled in these ants". However, I miss some figure clearly supporting this statement.

I feel that this question (specialization VS experience) has the potential to draw the attention of wide readership but, unfortunately, in its current state, evidence is not strong enough to be really convincing. Indeed, conclusions are indirectly drawn from statistical analyses using a multi-model approach and collinearity adjustments. The evidence is too indirect and needs to be more straightforwardly supported by experimental data.

On existing data, the authors should first show on a new figure how the level of specialization (proportion of baseline tandem runs in which an ant played a given role) relates (or not) to experience (the number of baseline tandem runs in which an ant played a given role).

More importantly, new experiments should be done in order to decouple highly specialized individuals (ideally with little experience by having participating to only a few tandem runs) from experienced ones (elite individuals performing many runs but with no clear-cut specialization). Further experiments should be designed either to "create" or at least to identify elites in order to assess how their removal will alter the performance of tandem runs (in comparison to the removal of specialist individuals).

Discussion:

Some comments made on the introduction are also valid for the discussion

As regards the existence of highly specialized leaders VS the lower specialization of followers, this could be discussed in the light of the spatial location of individuals.

Indeed, the probability of being a leader is closely linked to the spatial location of one individual outside the nest making it more likely to discover a new nesting site and hence to initiate a tandem run. Several individual traits such as older-age make this subset of individuals more eager to stay in the outside (and hence more "specialized" in external tasks such as in the leading of tandem).

By contrast, any colony member stay inside the nest for some amount of time and hence any colony member can potentially become a "follower" during nest moving. This may account for the higher and lower "specialization" of leaders and followers respectively .

Minor comments:

Why using two new nesting sites (instead of a single one)? Please-justify the choice of this experimental setup.

Line 39: "generalist `elite' workers that are highly active across a broad array of tasks and thus accrue high experience without specialising upon particular tasks." The cited reference n°19 of "Charbonneau et al 2017" is not appropriate here (no elites reported in this work). Please also see the paper of Robson S.K., Traniello J.F.A. (1999) Key individuals and the organisation of labor in ants. In: Detrain C., Deneubourg J.L., Pasteels J.M. (eds) Information Processing in Social Insects. Birkhäuser, Basel.

Reviewer #1 (Remarks to the Author):

Manuscript Number: COMMSBIO-20-2367

Title: Leadership -- not followership -- determines performance in ant teams

The study used tandem running in *Temnothorax* ants to investigate division of labour in ants. This approach allowed to measure performance (successful relocation, distance to target nest) and to compare it between groups in which either the most active leaders, followers, both or hardly specialised ants were removed. To do so, a baseline for each ant was assessed in 4 emigration trials, before the manipulated 5th trial took place.

The study was well done, the manuscript is well written and clearly conveys the hypothesis and the used procedures. The statistical analysis is thorough and well documented. All in all, I have found no major issues and approve the publication in *Communications Biology* after the questions below have been addressed:

Thank you very much for these comments. We are pleased that the reviewer found our study to be well done, and the paper clear and well written.

- The two measurements, success rate and distance covered, are not independent. All successful relocations always have a distance of 35cm. I would assume that the skewed distributions (many 35s; as shown in figure 1c) led to a violation of normality of residuals required for the LMM. Can the authors disprove that claim?

Thank you for pointing out that the success rate and the distance covered had some degree of overlap. We have now re-defined the performance metrics to reduce such redundancies. We still consider the success rate across all tandem runs, giving a first coarse measure of performance. We then analyse the straight-line distance *only* for tandem runs that did not progress all the way to the new nest, providing a more refined measure of the performance of failed tandem runs. Furthermore, we introduce a new metric providing an additional measure of performance for successful tandem runs only: the time taken by the tandem run to reach the new nest.

Quantile plots for the residuals from the mixed models testing for between-treatment performance differences (presented in Fig. 3c-e & described on L. 110-114) are shown below. Shapiro-Wilks tests on these distributions produced non-significant results for all three performance metrics. Hence, our mixed models do not violate the assumption of residual normality.

Figure R1. Residual quantile-quantile plots for the colony-level mixed models testing for an effect of treatment upon tandem performance. The residuals are normally distributed.

- Distance covered suggests that this is an estimate of how many centimetres the ants travelled. However, any sinuosity of the path is not reflected in the measurement. Can the authors approve that a straight line between the nests is a good approximation of the travelled route? Otherwise, the name of the metric is

misleading.

The reviewer is correct, our metric description was confusing. We did not intend for our metric to stand as an approximation for total distance travelled, but rather to measure how close to the new nest the tandem run had brought the follower.

To clarify this we have now renamed the metric to ‘straight-line distance’.

• Moreover, the measures established in the manuscript might not capture follower performance in its entirety. Poor followers might cause detours or longer breaks. This might be reflected by the time taken per trip. Can the authors provide any insights on travel times?

Yes. In the new version of the manuscript we have added a new performance metric; the time taken to travel from the initial nest to the target nest, which we defined for successful tandem runs only (Fig. 1e, Fig. 3e, Fig 4c,f). Overall, we found no significant predictors of the time successful tandem runs take to reach the new nest, but we included it in the manuscript for completeness.

• While we would expect a leader ant to do multiple trips per trial, it is interesting that also some followers do multiple runs within one trial. Probable explanations for that behaviour could be added to the discussion from line 162. In case that followers also have important functions such as e.g. cross-comparisons of nests known by previous research, it should be added as information to the introduction.

Thank you for this suggestion. It is true that recruits that act as a follower in multiple tandem runs could be doing so to make comparisons between nests. This is certainly a very interesting hypothesis, and there have been a number of studies on this in *Temnothorax* ants (Robinson et al. 2011), and also in the honeybee (Visscher & Camazine, 1999). So far there has been no evidence that direct comparisons by individual ants play any significant role in nest site selection (Robinson et al 2009 Proc B, Robinson et al 2014 Proc B). A more likely explanation for multiple tandem following by specific ants is that they were not convinced that the nest was good enough upon the first visit and did not commit to it, thus remaining available to be recruited in subsequent tandem runs. However, as to the best of our knowledge there is no evidence from previous publication that followers have important functions such as cross-comparisons of nests, we have not added to the introduction.

• The authors included both S_{lead} and S_{follow} in models during the model selection procedure. However, to my understanding, $S_{lead} = 1 - S_{follow}$. This information would thus be redundant, and these predictors are not independent. I am thus unsure about the validity of such a model. Can the authors exclude an effect on the model selection process? E.g. S_{lead} was not always in the model before S_{follow} was added.

Thank you. In fact, in our original manuscript S_{lead} represented the specialisation of the leader (in leading) and S_{follow} the specialisation of the follower (in following). These measures are totally independent from one another, as they apply to two different individuals, and it was not the case that $S_{lead} = 1 - S_{follow}$. In any case, in light of the reviewer 2’s comments, we have now re-run the individual-level analyses without the specialisation metrics. This greatly simplified the analysis, which now consists of only four main effects (and their interactions), namely;

- Leading activity, A_L - the number of tandem runs in which the focal leader had previously acted as a leader.

- Following activity, A_F - the number of tandem runs in which the focal follower had previously acted as a follower.

- Leading consistency, C_L - the number of previous emigrations in which the focal leader had acted as a leader at least once.

- Following consistency, C_F - the number of previous emigrations in which the focal follower had previously acted as a follower at least once.

(Note, as both ‘emigration experience’ and ‘tandem experience’ were both aspects of experience, we have re-named them as ‘consistency’ and ‘activity’ respectively, and have mentioned this re-labelling on L68 & L144 of the main text.)

As in our previous analysis, this simplified individual-level analysis still finds that the leader is the key determinant of tandem performance (see Fig. 4).

Regarding the reviewer’s last point about model selection, and the possible effects of the order in which terms are added; instead of evaluating the significance of all terms in the full model (which can indeed be affected by term ordering), the multi-model selection approach consists in considering all possible models and sub-models and calculating the AIC of these models as a whole, hence ordering is not an issue. This can be shown without doubt as the model selection produces exactly the same results regardless of the order in which the terms are entered for the full model.

Minor

• In Table 1 I would suggest to sort by species

Thank you. we have now removed Table 1 as we no longer explicitly consider specialisation.

• In line 95, MSE refers to mean squared error or SEM?

This was indeed an error – it should have been SEM. We have corrected it.

• I would suggest to add a small table or inset to figure 2 where the composition/procedure for the four groups is listed (lines 290 – 300)

Thank you very much for this suggestion. We have now extended Fig. 3 (originally Figure 2) to illustrate the targeted removal of a prominent leader (Fig. 3a), and to show the effect of the targeted removals on the composition of tandem runs in terms of the activity and consistency of leaders and followers across the four treatments (Fig. 3b). Furthermore, we have also extended Fig. 3 to illustrate the between-treatment contrasts for the three performance measures (Fig. 3c-e).

• Very long sentence starting at l178, split.

We agree with the reviewer. With the re-organisation of the Discussion, this sentence has now been replaced.

• Line 337, so here no removal of reversal runs and switches took place?

Thank you. All our analyses excluded reverse tandems and tandems involving follower switching. This is stated in the Methods (L. 324).

• I would suggest to add the average baseline counts for the variables specialisation, emigration experience and tandem experience in the main text, so people do not have to extract them from figure S6.

Following the reviewer’s suggestion, we have extended Fig. 3 (originally Fig. 2) to include a graphic in which the composition of tandem runs in the four targeted removal treatments are shown as colour-coded leaders and followers. We believe that this visual depiction is the best way to illustrate to readers the pair composition differences between the treatments, and we thank the reviewer for suggesting it.

• Line 56 exchange 'brunt' with 'most' to make the manuscript more accessible

We have removed ‘brunt’.

• L75 lost contact

We have simplified the sentence to, “Second, task performance can be readily measured, as the tandem run is successful if the leader guides the follower all the way into the new nest, but unsuccessful if the pair lose contact with one another en route”. We kept the present tense as this is a general statement that applies to all tandem runs in general rather than only those we recorded in our experiments.

• Fig1 the corresponding randomly-selected

Thank you. In the new version of the paper, the recruitment networks are shown in Fig. 3, hence the legend has been modified and no longer includes this statement.

- L179 because of the high degree

With the re-organisation of the Discussion, this sentence has now been replaced.

- L279,80 sentence is confusing, maybe write: two ranks, one for the leading and one for the following task for each individual

The reviewer is correct that this was confusing. We have rewritten the sentence to read, “To identify prominent leaders and followers we used a hierarchical ranking procedure to assign two ranks to each individual, one for the leading task and one for the following task.”

- L319, shouldn't it be replicate nested in colony identity?

Our experiment consisted of three replicates which were run at different times of year, and four colonies per replicate. Because different colonies were used in different replicates, replicate is not nested within colony but colony is nested within replicate.

We describe the experimental structure in the first paragraph of the Materials and Methods (L. 227-232). Further details about the colony collection dates, colony demographics, and duration of each replicate are provided in Table S1, and on L. 362-366 we have elaborated the description of the nesting structure for the random effects in the individual-level models.

Reviewer #2 (Remarks to the Author):

Division of labor is often mentioned as one of the factors (among others) responsible for the ecological success of ants. Increased task performance at the colony level usually outcomes from a higher tasks' achievement by specialized individuals.

This research work is organized around two complementary axes.

The first one aims to specify the relative importance of the ants specialized in the leading and in the following on the effectiveness of tandem runs . The second axis aims to understand through statistical models whether task specialization or experience of the observed individuals are the main factors influencing the performance of tandem runs.

The paper is very well-written, figures in the main paper are informative and supported by many useful tables/figures in the supplementary.

We are pleased that the reviewer found our paper informative, and we would like to thank the reviewer very much indeed for their detailed comments and constructive feedback.

Here follow my main comments on the different sections of this paper

Introduction

The authors underline that a positive association between specialization and task performance is relatively little supported by existing literature. I agree with them as far as experimental work on monomorphic workers is concerned. This is far less true when considering the many papers that demonstrated the link between task performance and specialization in polymorphic ants as well as by the theoretical studies made on this topics.

Here follow some suggested references (far to be an exhaustive list...)

-Wilson EO. Caste and division of labor in leaf-cutter ants (Hymenoptera: Formicidae: Atta). Behav Ecol Sociobiol. 1983; 14:

- Wilson EO. The relation between caste ratios and division of labor in the ant genus *Pheidole*. *Behav Ecol Sociobiol.* 1984
- Detrain, C., & Pasteels, J. M. 1991. Caste differences in behavioral thresholds as a basis for polyethism during food recruitment in the ant, *Pheidole pallidula* J. *Insect behave.* 4(2)
- Mirenda JT, Vinson SB. Division of labour and specification of castes in the red imported fire ant *Solenopsis invicta* Buren. *Anim Behav.* 1981; 29.
- Johnson BR. Organization of work in the honeybee: a compromise between division of labour and behavioural flexibility. *Proc R Soc Lond B Biol Sci.* 2003
- Theraulaz, G., Bonabeau, E., & Deneubourg, J. N. (1998). Response threshold reinforcements and division of labour in insect societies. *Proc Royal Soc London. Series B: Biological Sciences*, 265(1393).
- Also many studies on Army ants

We agree with the Reviewer that there is much better evidence for a link between performance and specialisation in polymorphic species than in monomorphic species, and we have added a sentence in the Introduction to make this clear “... although there is strong evidence for a link between division of labour and performance in humans and the minority of social insect species with polymorphic worker castes (Wilson, 1980; Mirenda & Vinson, 1981; Detrain & Pasteels, 1991; Johnson, 2003), there is little evidence for such a link the majority of species that have monomorphic workers.”

To illustrate division of labor, the authors chose to study cooperative tandem runs during nest emigration in *Temnothorax* ants. I wonder if the leading “task” and the follower “task” can be considered as a division of labor similar to that observed between e.g. nest cleaners, nurses or foragers. This appears to me rather as a single task based on a communication comparable to that of other recruitments (by group or by trail). At least, the authors should comment on this and widen their conclusions/discussion on the influence of leaders (scouts) VS followers in group and trail recruitment.

Thank you for these perspectives. We believe that the leading “task” and the follower “task” can be considered as a true division of labour for two reasons. First, leading and following actually involve quite distinct actions, and appear to play different functional roles. For example, leaders must retrieve their memories to navigate the way back to the new nest whilst also waiting for and evaluating the tactile feedback provided by the follower, whereas followers must learn the route, whilst providing tactile feedback to the leader.

Second, our new data and previous studies show that tandem leaders typically have a higher leading consistency than their followers, whereas followers have a higher following consistency than their leaders (Fig. S5 b). There was also a significant negative correlation between the number of tandem runs an individual leads in one emigration, and the number it follows in subsequent emigrations (Fig S4 c,f). Even though the division of labour between tandem leaders and followers may be less conspicuous than that between nurses and foragers, these results are highly suggestive of a stable division of labour between specialist leaders and specialist followers. Originally, we did not detail these results in great depth in the main paper because we had previously documented these correlations in the tandem runs of another species of *Temnothorax* ant (Richardson *et al.*, 2018). Still, we have now fleshed out the first section of the Results, where we detail the evidence for a division of labour between leaders and followers (L. 85-105).

To better understand the origin of the division of labour between leaders and followers, we have carried out a new analysis to quantify the propensity of different groups of workers to switch between leading and following within a given emigration. This analysis identified two clear switching biases; consistent leaders are more likely to switch to leading after having followed, and consistent followers are more likely to switch to following after having led. This suggests that the division of labour between leading and following specialists is maintained by asymmetric switching preferences, with individuals preferring to switch towards the task that they are specialised upon. As this new analysis helps to clarify why the leading ‘task’ and the follower ‘task’ can be considered a real division of labour, we have added a new section in the main text (L. 106-118), and included a new main text figure (new Fig. 2). We have also added a description of the procedures in the Methods section, ‘Permutation tests for task-switching’. We would like to thank the reviewer for pushing us to investigate this more deeply.

Finally, as we believe that many more subtle or fine-grained divisions of labour likely remain undiscovered in nature, we have rewritten the Discussion to open as follows;

“Historically, most work on social insect division of labour has focused on the most conspicuous examples, such as between reproductive and sterile worker castes (Volny & Gordon, 2002), between polymorphic worker castes (Oster & Wilson, 1978), and between ‘temporal castes’ of younger and older workers (Seeley, 1982). However, although difficult to identify, more subtle forms of divisions of labour appear also to be ubiquitous. For example, whereas the division between inside- and outside nest workers is found in nearly every species of social insect, among inside-nest workers of the pharaoh ant, there is a further subdivision between workers that specialise on feeding either young or old larva (Walsh et al., 2018). Similarly, in outside-nest workers of the honeybee foraging is itself subdivided between scouts that explore for new resources, and recruits that exploit them (Seeley, 1983).”

Removal Experiments

The first section reports experiments assessing how much the removal of either top-specialized leaders or followers will influence the performance of tandem runs. The authors found that the removal of specialized leaders had more impact than that of followers as shown by figure 2.

I am convinced about the influence of prominent leaders on the quality of tandem runs. However, the sample size associated to graph bars and statistical tests is not given. The figure caption specifies that the average values were calculated on all tandem runs but the sample size should be indicated over each bar.

We are pleased that our experiments and analysis convinced the reviewer of our central result; that it is the leaders that are the key to tandem run performance.

We have now added the sample sizes next to the bars in Figure 2.

Furthermore, I think that it is not correct to average values measured on the individual tandem runs. As a rule, if one wants to evaluate the impact of a manipulation (here the removal of a subset of workers out of a given colony), the measure of this impact should also be done at the colony level. This means that tandem run performance should be done for each of the tested colony (e.g. the average distance of all tandem runs performed during one nest emigration by a given colony). This also means that bars shown on figure should be averaged on a sample of only $n=3$ (knowing that each of the 12 colonies is assigned to one of four conditions for the 5th nest emigration).

Thank you for these comments. As the reviewer suggested, in our revised paper we have replaced the individual-level analysis with a colony-level analyses in which each data point is a within-colony mean. Despite this change our results remain the same: we still find that the significant performance reductions are only observed in the targeted removal treatments that involved the removal of specialist leaders (Fig. 2C,d). The consistency in the results of the individual- versus colony-level analysis is because our original individual-level mixed effect models had specified colony identity as a random effect.

Regarding the averaging of the bars in Figure 3; the bars did (and still do) show the grand means, calculated from the colony means, as the reviewer suggests. We have added a sentence to make that clear in the caption.

Such a sample size is definitely too small and needs to be increased to check whether the reported differences would be maintained. I am well aware that this requires additional experiments but, on the other hand, such a colony-level analysis will provide additional interesting data such as the possibility to compare the dynamics of tandem runs along the emigration process across the different conditions of workers' removal.

Thank you very much for these thoughtful comments. The reviewer's objections about sample sizes seem to stem from concerns that our 12-colony experimental design might suffer from low statistical

power and that the significant effects reported, on which our paper’s main conclusions rely, could disappear if the sample size was increased.

While we would have loved to increase the sample size by performing more replicates, this was not feasible from a logistic point of view due to the sheer amount of time involved in carrying out each replicate; as such our experiments already took 9 months full-time work by a Master’s student (AC) who has unfortunately left the lab since, as have the other two non-senior authors in the paper (NS and TR).

To test whether our sample size is indeed too low, as the Reviewer fears, we carried out formal power analysis on the two main significant analyses using the R package SIMR (Green & MacLeod, 2016). This procedure assesses the statistical power of a given experiment to detect a given effect size with a given level of statistical significance. For this analysis, we used the effect size estimates provided by the new colony-level models, and we set the significance threshold to the standard level of 0.05.

This analysis revealed that our experiment had a power of $82 \pm 2\%$ (mean & 95% confidence interval from 1000 Monte Carlo simulations) for the mixed model testing how tandem success rate depends upon the removal treatment (Fig. 3c). The power was $90 \pm 2\%$ for the straight-line distance model (Fig. 3d). These values are greater than the typically recommended minimum statistical power of 80% (Murphy et al. 2014), and significantly greater than the typical power of papers in the field of Animal Behaviour and Behavioural Ecology (Jennions & Møller, 2002; Fig. R2).

Figure R2. Comparison of the power for the models presented in the text, with the powers for tests reported in 10 other journals in the field. Dashed lines & shaded areas represent the mean power and its 95% confidence interval for the two models showing a significant influence of treatment upon performance. The white points show the mean power for the ten journals reported in Table 2 in Jennions & Møller (2002).

Specialization VS Experience

The second section aims to disentangle the effects of experience and specialization on the performance of tandem runs.

At line 79, the authors state that they “capitalized upon the observation that specialisation and experience are only loosely coupled in these ants”. However, I miss some figure clearly supporting this statement.

I feel that this question (specialization VS experience) has the potential to draw the attention of wide readership but, unfortunately, in its current state, evidence is not strong enough to be really convincing. Indeed, conclusions are indirectly drawn from statistical analyses using a multi-model approach and collinearity adjustments. The evidence is too indirect and needs to be more straightforwardly supported by experimental data.

On existing data, the authors should first show on a new figure how the level of specialization (proportion of baseline tandem runs in which an ant played a given role) relates (or not) to experience (the number of baseline tandem runs in which an ant played a given role).

Thank you for these very thoughtful comments.

Given the reviewer's misgivings and as these specialisation metrics were indeed highly correlated with the experience in leading and following, we have now completely dropped the specialisation ratios from the multi-model analysis. This means that the multi-model analysis is now much simpler, as it now consists of only five one-way main effects, namely;

- Tandem rank – the rank order of appearance of a given tandem run within a given emigration.
- Leading activity, A_L - the number of tandem runs in which an individual acted as a leader.
- Following activity, A_F - the number of tandem runs in which an individual acted as a follower.
- Leading consistency, C_L - the number of emigrations in which an individual acted as a leader at least once.
- Following consistency, C_F - the number of emigrations in which an individual acted as a follower at least once.

(Note, as both 'emigration experience' and 'tandem experience' were both actually aspects of experience, we have re-named them as 'consistency' and 'activity' respectively, and have mentioned this re-labelling on L68 & L144 of the main text.)

This simplified analysis still finds that the leader, rather than the follower, is the key determinant of tandem performance (see Fig. 4).

On the question of whether or not the evidence is strong enough to be really convincing, we would like to point out that in addition to the high statistical power for of the new colony-level models that we presented in reply to the reviewer's previous point, the power for the individual-level analysis of overall tandem success rate was $88 \pm 2\%$ (using the effect size for C_L of 0.38 estimated by the model-averaging & a significance level of $p < 0.05$). This figure is again above the recommended statistical power of 80%, and hence we suggest that the evidence is actually both strong and convincing.

More importantly, new experiments should be done in order to decouple highly specialized individuals (ideally with little experience by having participating to only a few tandem runs) from experienced ones (elite individuals performing many runs but with no clear-cut specialization). Further experiments should be designed either to "create" or at least to identify elites in order to assess how their removal will alter the performance of tandem runs (in comparison to the removal of specialist individuals).

We agree that to unequivocally map the causal links between performance, specialisation and experience, would require manipulative experiments that 'create' workers of all kinds (e.g. high leading experience, low leading specialisation, etc), and then directly control the allocation of particular individuals to particular tasks. Unfortunately, this degree of manipulative control would be very challenging to achieve, even for tasks that one individual can carry out alone, but it would be even more challenging for cooperative tasks with defined roles, such as tandem running.

Although in our experiments we did not attempt to 'create' individuals with particular characteristics, or directly control exactly which individuals performed which tasks in the tandem run, our targeted removals did successfully alter tandem running composition by fostering the formation of atypical tandem runs, with leaders and followers performing tasks that they normally would not attempt. Further, these perturbations of the normal allocation of workers to tasks were successful in inducing significant performance reductions. Moreover, most studies in the field have been purely descriptive,

and even among the manipulative studies, none have attempted to use repeated whole-colony emigrations to quantify the roles of over a thousand individually-marked workers, and certainly no other studies have attempted to perform such highly controlled targeted removals to quantify the role of prominent individuals within the colony. Hence, although certainly not the last word on the subject, our work has developed novel methods and generated novel results, and represents a significant advance in the field.

Lastly, since the new version of the manuscript does not attempt to tease apart specialisation and experience, we feel that further experiments to separate their influence are no longer necessary.

Discussion:

Some comments made on the introduction are also valid for the discussion.

As regards the existence of highly specialized leaders VS the lower specialization of followers, this could be discussed in the light of the spatial location of individuals.

Indeed, the probability of being a leader is closely linked to the spatial location of one individual outside the nest making it more likely to discover a new nesting site and hence to initiate a tandem run. Several individual traits such as older-age make this subset of individuals more eager to stay in the outside (and hence more “specialized” in external tasks such as in the leading of tandem).

By contrast, any colony member stay inside the nest for some amount of time and hence any colony member can potentially become a “ follower ”during nest moving. This may account for the higher and lower “specialization” of leaders and followers respectively .

We agree with the reviewer that task specialisation is associated with worker age (e.g., through temporal polyethism), and that workers that specialise upon a particular role will also exhibit a spatial fidelity pattern indicative of the tasks that comprise their role. Therefore, we have rewritten the last paragraph in our new discussion to highlight that the tendency to lead has been associated with a range of different factors, including age and spatial position within the nest (L. 215).

Minor comments:

Why using two new nesting sites (instead of a single one)? Please-justify the choice of this experimental setup.

We used a binary choice experimental design because we found that presenting a colony housed in a low quality nest with a choice between two identical high quality nests elicits greater numbers of tandem runs.

To better explain the rationale for the design we have added a sentence to the ‘Colony Emigrations’ section in the Methods; “*This design was chosen because the quality difference between the initial and new nests motivates nest-site discoverers to lead tandem runs, whilst the binary choice between identical nests delays the formation of a consensus, leading to large numbers of tandem runs (Dornhaus et al. 2004, Richardson et al. 2018).*”

Line 39: “generalist `elite' workers that are highly active across a broad array of tasks and thus accrue high experience without specialising upon particular tasks.” The cited reference n°19 of “Charbonneau et al 2017” is not appropriate here (no elites reported in this work). Please also see the paper of ‘Robson S.K., Traniello J.F.A. (1999) Key individuals and the organisation of labor in ants. In: Detrain C., Deneubourg J.L., Pasteels J.M. (eds) Information Processing in Social Insects. Birkhäuser, Basel.

The reviewer is correct about Charbonneau. Since we have rewritten the paper to avoid explicitly measuring specialisation, this sentence has been removed. Nevertheless, both Charbonneau et al (2017) and Robson & Traniello (1999) are cited in the new Introduction.

References

- Green, P., & MacLeod, C. J. (2016). SIMR: an R package for power analysis of generalized linear mixed models by simulation. *Methods in Ecology and Evolution*, 7(4), 493-498.
- Jennions, M. D., & Møller, A. P. (2003). A survey of the statistical power of research in behavioral ecology and animal behavior. *Behavioral Ecology*, 14(3), 438-445.
- Murphy, K.R., Myers, B. & Wolach, A. (2014). *Statistical Power Analysis: A Simple and General Model for Traditional and Modern Hypothesis Tests*, Fourth Edition. AOCS
- Robinson, E. J., Franks, N. R., Ellis, S., Okuda, S., & Marshall, J. A. (2011). A simple threshold rule is sufficient to explain sophisticated collective decision-making. *PloS one*, 6(5), e19981.
- Visscher, P. K., & Camazine, S. (1999). Collective decisions and cognition in bees. *Nature*, 397(6718), 400-400.

REVIEWERS' COMMENTS:

Reviewer #1 (Remarks to the Author):

I am satisfied with the answer to my questions and how my suggested changes were implemented. The manuscript underwent a major revision, most notably the new multi-modal approach using consistency and activity, which is improving both the analysis and the readability.

Overall, I only have a few minor comments:

L155 you mention importance, but never explain its meaning outside of the supplements. To aid understanding, you might want to add a short explanation, e.g. from the supplement's text: "a predictor that appeared in all models or many of the high-ranked models would receive a relative importance close to 1 and close to 0 if it appeared in the least likely models"

L354 you might want to provide the three metrics here again

L355-365 the multi-model approach should be mentioned here shortly, with reference to the supplements, so it is clearer to the reader.

L80 followers

L349 artifacts

L362 replicate

L366 remove model

Reviewer #2 (Remarks to the Author):

The authors have made a substantial effort to improve the MS and to meet my comments. Thank you!

I have really appreciated their answer to the concept of « task specialisation » as well as their supplementary analysis of asymmetric switching between leading and following activity.

However, in my opinion, there is still an issue about the limited sample size used to assess the impact of leader or follower removal on colony-level emigration performance. As stated in fig 3 caption, means & errors bars in figure 3 represent the grand means calculated from the colony means. So if I understand well, these means and errors are each calculated on a sample size of $n=3$ colonies per condition. Likewise, I guess that colony-level analysis of the three performance metrics (success rate, straight line distance, time taken) are also carried out with a sample size of 3 per condition.

The authors made a formal power analysis using the R package SIMR to detect a given effect size with a given level of statistical significance that seems to be reassuring from a mathematical perspective. However, one may wonder about the validity of such a power analysis (I am not familiar with) when being based on such a small sample. This has to be checked by a statistician. In any case, this sample size is lower than what is usually found in the field of behavioural sciences. In my opinion, this prevents to currently draw conclusions at the colony-level. Of course, my remark does not invalidate the individual -level analyses.

Minor comments :

If the bars on figures 3c-e are actually the means calculated from the colony means, then the n value in parentheses are somewhat misleading. They should all be equal to $n=3$

On figure 1B, I would suggest to reverse the ordering of ticks on the Y axis so as to be in parallel to the measure of the straight-line distance